# An Experimental Study on Biochar/Polypyrrole Coating for Blade Anti-Icing of Wind Turbines

Xiaoheng Li [1], Xiaojuan Li [1], Zhongqiu Mu [1], Yan Li [1,2,*] and Fang Feng [2,3,*]

1 College of Engineering, Northeast Agricultural University, Harbin 150030, China
2 Key Laboratory of Icing and Anti/De-Icing, China Aerodynamics Research and Development Center, Mianyang 621000, China
3 College of Arts and Sciences, Northeast Agricultural University, Harbin 150030, China
* Correspondence: liyanneau@neau.edu.cn (Y.L.); fengfang@neau.edu.cn (F.F.)

**Abstract:** Wind turbines operating in cold regions are prone to freezing in winter, which can affect their performance and safety. To resolve this situation, the development of blade anti-icing technology has attracted widespread attention. In this study, a type of biochar/polypyrrole coating was obtained through synthesis on the surface of biochar. After characterization, it was found that the porous structure, irregular dents, and bumps on the surface of biochar/polypyrrole material contributed to the formation of a nanoscale roughness structure with a typical super-hydrophobic nanostructure. Additionally, it had a sufficient surface area. The wetting characteristics of the coating were analyzed with the assistance of a contact angle measurement instrument. The contact angle of the coating was determined as 151°, which indicates the excellent hydrophobic properties of the coating. Icing wind tunnel tests were carried out to evaluate the anti-icing effect of biochar coating and biochar/polypyrrole coating at different ambient temperatures and wind speeds. Compared with uncoated leaves, the icing area of biochar/polypyrrole coating was reduced. Additionally, the anti-icing effect of biochar/polypyrrole coating was most significant. This study provides a practical reference for the research of anti-icing coating on wind turbine blades.

**Keywords:** wind turbine blades; anti-icing coating; biochar; polypyrrole; icing wind tunnel test





## 1. Introduction

In the modern world, wind energy is mostly utilized for power generation. Since wind energy is inexhaustible and environmentally friendly, the utilization of wind energy has also attracted more and more attention in recent years. The weight of wind turbine blades tends to increase due to icing, which not only affects their operating efficiency but also puts the blades at risk of damage [1,2]. At present, the mainstream de-icing and anti-icing technologies can be divided to include active anti-icing and passive anti-icing ones. The active de-icing method aims to melt the ice through heating or mechanical measures, with anti-icing liquid sprayed onto the surface [3,4]. However, these methods are constrained by such problems as high cost and environmental pollution. The research on passive de-icing methods focuses mainly on the application of special coatings [5]. Due to the latest progress made in the development of nanotechnology and material engineering, nano-coatings have emerged as a promising solution to the prevention of icing and the promotion of de-icing [6,7], which makes them the focus of attention for research.

The water droplets existing on the super-hydrophobic surface tend to form spherical water droplets. Given the limited contact area between the water droplets and the super-hydrophobic surface, any small inclination angle can cause the water droplets to slip off the surface of the object instantly. The preparation process of super-hydrophobic ice-phobic coating is based on the following aspects. Firstly, by modulating the chemical properties and structural design of the surface of the material, the interaction at the solid-liquid interface is reduced so that water droplets are actively removed from the blade

surface before freezing, relying on other factors such as gravity [8]. Secondly, the anti-icing effect can be obtained by inhibiting the nucleation and growth of ice through the action of molecules and ions on the coating surface during the process of water droplet formation into ice [9]. After ice formation, the tiny air cushions formed between the rough surface structure of the ice-phobic coating reduce the contact area between the solid and the ice, acting as insulation and micro-cracks, which can effectively reduce the adhesion of the ice to the solid surface [10,11]. Xu [12] drew a comparison between the silicon–acrylic resin-coated blade and the uncoated blade to find out that the coated blade clearly outperformed the uncoated blade in anti-icing effect. Xi Zhang et al. [13] adopted electrochemical methods to deposit a layer of gold nanoparticles with dendritic nanostructures on the surface of polyelectrolyte-modified indium tin oxide electrodes. After the surface of gold nanoparticles was modified with dodecyl mercaptan, the contact angle of the composite interface reached 156°. Tarquini et al. [14] used super-hydrophobic materials to experiment on the ice coating of helicopter wings, revealing that the ice formed on super-hydrophobic surfaces was thinner and had lower adhesion strength than on normal surfaces. Despite the significantly improved hydrophobicity of the super-hydrophobic coatings, as discussed above, a simple static test is inadequate to meet the performance of the coatings and the anti-icing performance required in complex working environments. Through the icing wind tunnel experiment, a better understanding was gained as to the anti-icing performance of super-hydrophobic coating on wind turbine blades, the anti-icing effect of the hydrophobic coating surface was explored from a macro perspective, and the icing conditions of the coating surface were compared, which provides a practical reference for the research and development of anti-icing methods.

There are many super-hydrophobic materials whose application is limited by the high cost of synthesis. Biochar extracted from biogas residues is abundant and renewable. As a product of high-temperature bio-organic pyrolysis, biochar is affected by the rate and temperature of pyrolysis in terms of structure and properties. Due to the unique physical and chemical properties of biochar itself, the recombination with hydrophobic groups has great potential to facilitate the study of hydrophobic surfaces. Biomass resources can be carbonized and activated to produce biochar materials with micro- and nanostructures, which provides a basis for the preparation of biochar-based hydrophobic materials. The research of DIEDKOVA K [15] on biodegradable nanofiber scaffolds has aroused the interest of the author. Polypyrrole has now attracted increasing attention due to its excellent film-forming property, high stability, and ease of synthesis [16]. Additionally, polypyrrole possesses a rough micro-nano hierarchical structure, which is conducive to significantly reducing the contact area between materials and water and reducing the adhesion of ice. This is essential for the smooth removal of ice with external mechanical forces. Therefore, it is considered a promising type of hydrophobic material.

Previous studies on the preparation of the coating operation were more complex and not energy-efficient. In addition, there were few studies on the anti-icing performance of coated blades. In this study, biochar/polypyrrole coatings with a simple preparation process were prepared using digested waste from anaerobic fermentation. The materials were characterized using Fourier transform infrared spectroscopy (FTIR), X-ray Diffraction (XRD), Search Engine Marketing (SEM), and Brunauer Emmett Teller (BET) measurement instruments. The hydrophobicity was measured by contact and roll angles. The icing strength of uncoated and coated was measured by a dynamometer. The anti-icing effect of the coatings was analyzed by icing wind tunnel experiments at different wind speeds and temperatures. The results of this study help to provide a new strategy for the resource utilization and rational consumption of digested waste from anaerobic fermentation and provide new ideas to be applied to the anti-icing of wind turbines in practice.

## 2. Materials and Methods

### 2.1. Coating Materials and Preparation

Table 1 lists the main reagents used in the test and their corresponding manufacturers. The inoculum of cow dung was prepared with the total solids (TS) set to 7%. After 21 days of fermentation, the bottom sediment was removed and dried by heating it to 600 °C for 60 min in a tubular furnace with argon as the protective gas. Then, it was naturally cooled to obtain biogas residue biochar. After being dispersed in 300 mL of HCl solution (1.0 mol/L), the obtained biochar was added with 0.1 mL of pyrrole monomer, 0.1 mg of thioacetamide, 1.0 g of cetyltrimethylammonium bromide, and 2.02 g of ammonium persulfate, respectively. The biochar/polypyrrole material was synthesized by chemical oxidation method, using ammonium persulfate as oxidant, oxidation reaction with pyrrole monomer, hydrochloric acid solution as a solvent, and cetyltrimethylammonium bromide as a surfactant; the observed change in color from white to partial black to completely black over time. The above reagents were analytical reagent grade. The polymerization reaction was allowed to proceed for 12 h. The product was centrifuged and vacuum dried to obtain biochar and polypyrrole, grinding the product. Disperse PDMS and curing agent in 25 mL of N-Hexane in the ratio of 10:1, take 1 g of the same mass as PDMS and mix it, sonicate for 0.5 h, and then stir for 0.5 h, then pour it into the gun container, choose 0.2 MPa for gun pressure and keep 15 cm distance from the blade to spray evenly, put it into the oven for 2 h at 80 degrees Celsius to dry.

**Table 1.** Experimental reagents.

| Drug Name | Manufacturer |
| --- | --- |
| Hydrochloric acid (HCl) | Harbin Chemical Reagent Factory, Harbin, China |
| Pyrrole monomer | Shanghai Jingchun Reagent Co., Ltd., Shanghai, China |
| Thioacetamide | Shanghai Ziyi Reagent Factory, Shanghai, China |
| Cetyltrimethylammonium bromide | Shanghai Ziyi Reagent Factory, Shanghai, China |
| Ammonium persulfate | Tianjin Bodi Chemical Co., Ltd., Tianjin, China |
| Anhydrous ethanol | Tianjin Fuyu Fine Chemical Co., Ltd., Tianjin, China |
| Polydimethylsiloxane (PDMS) | Dow Corning Corporation, Auburn, MI, USA |
| N-Hexane | Tianjin Fuyu Fine Chemical Co., Ltd., Tianjin, China |
| Cow dung | Harbin Rear Xujiatun Cattle Farm, Harbin, China |

### 2.2. Characterization Method and Equipment

Sample preparation using the KBr tableting method, first mix the appropriate amount of powder to be measured and KBr crystal in the agate mortar and grind it into a very fine powder, and then put it into the mold, and can be tested into thin slices with the tablet press. Fourier transform infrared spectroscopy was used to analyze the characteristic vibration of functional groups or chemical bonds. After 100 °C, degassed for 5 h were tested using a specific surface area and porosity analyzer ASAP2460. The sample powder was fixed on the sample table with double-sided tape, and the glow was sprayed with gold for observation. The SEM was used to characterize the surface and internal micromorphology of the prepared material. The wettability of the sample surface was measured using an optical contact angle meter. In order to perform contact angle measurements, the acquisition mode of the instrument was selected as single acquisition analysis, the spiker syringe was fixed to the mounting table, the sample was placed on the sample table, and 5 µL of water was dropped onto the sample during the measurement, and the measured contact angle magnitude is realistically presented in the database. For rolling angle measurement, the instrument selects the rolling angle measurement mode and fixes the pointed syringe on the mounting table, and the sample is placed on the sample table. A total of 5 µL of water is dropped onto the sample during the measurement, and the rotating platform starts to rotate, and the platform movement stops automatically when the software detects the droplet movement, and the platform tilt angle is the rolling angle. The characterization test and equipment required for materials are detailed in Table 2.

**Table 2.** Test equipment.

| Instrument Name | Model | Manufacturer |
|---|---|---|
| Fourier infrared spectrometer | LPHA-T | Bruker, Ettlingen, Germany |
| Field emission scanning electron microscope | SU8010 | Hitachi, Tokyo, Japan |
| Specific surface area and porosity analyzer | ASAP2460 | Mack Corporation, Arlington, VT, USA |
| Contact angle measuring instrument | JCD2000D3M | Shanghai Zhongchen Digital Technic Apparatus CO, Ltd., Shanghai, China |
| Digital display push-pull gauge | HP-300 | Edberg Instrument Co Ltd., Beijing, China |

### 2.3. Anti-Icing Test System and Method

The anti-icing test of the icing wind tunnel was performed on the low-speed reflux wind tunnel of the Wind Energy Research Laboratory of Northeast Agricultural University. As shown in Figure 1a, it is a low-speed, return-flow wind tunnel. The refrigeration and spray devices were installed in normal wind tunnels to create a low-temperature icing environment [17]. The test section of the cryogenic icing wind tunnel was sized 250 mm × 250 mm. The wind speed ranged from 1 m/s to 20 m/s. The ambient temperature was restricted to the range of −20–0 °C. The blade airfoil chosen for the experiment was NACA0018, and the experimental material was glass fiber reinforced plastic composite, of which the chord length was 100 mm, and the span was 20 mm. The ambient temperature in July is 20 °C. Through different spray systems, The water spray pressure is 3 MPa, and the different liquid water contents (LWC) and diameters of supercooled water droplets (MVD) were obtained. LWC ranged from 0.1 g/m³ to 5 g/m³, while MVD ranged from 20 μm to 100 μm.

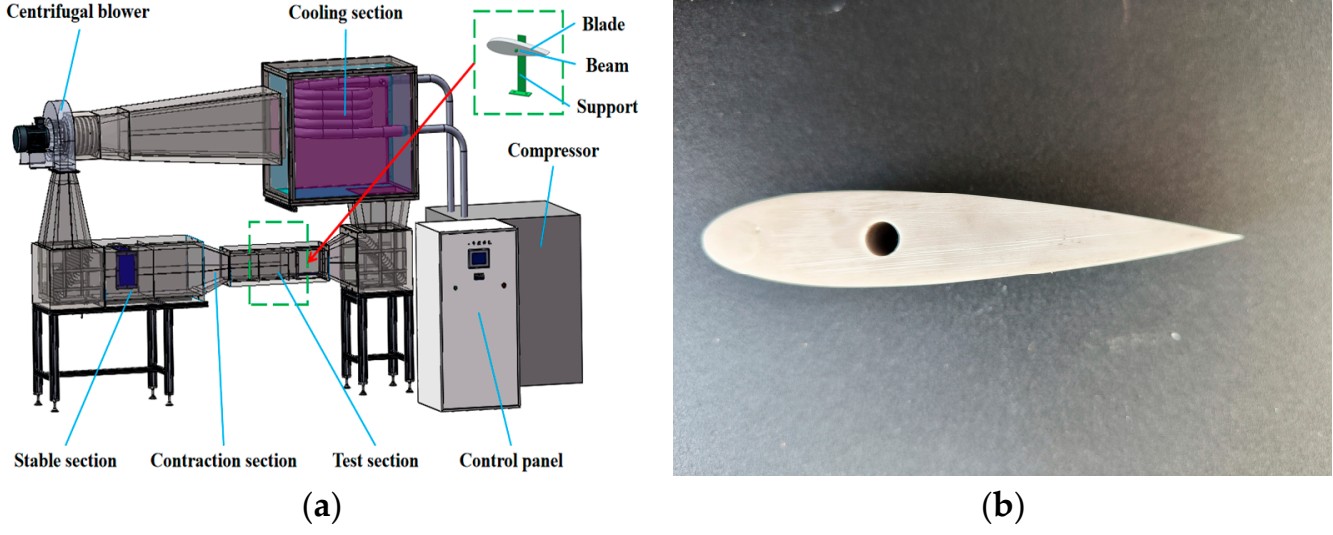

**Figure 1.** Schematic diagram of (**a**) icing wind tunnel test system and (**b**) test blade model.

The test blade was fixated on the test bench. After the wind tunnel was opened up, refrigeration was performed. The wind speed was set under nine different conditions, as shown in Table 3. With the start-up of the spray system, the supercooled air was mixed with the water mist in the test section of the wind tunnel to strike the surface of the supercooled test model. The change of surface icing was analyzed at an interval of 20 s.

**Table 3.** Test conditions of the icing wind tunnel system.

| Angel of Attack (Degree) | Wind Speed (m/s) | Temperature (°C) | Icing Time (min) |
|:---:|:---:|:---:|:---:|
| 0 | 5 | −5 | 0–3 |
| 0 | 5 | −10 | 0–3 |
| 0 | 5 | −15 | 0–3 |
| 0 | 10 | −5 | 0–3 |
| 0 | 10 | −10 | 0–3 |
| 0 | 10 | −15 | 0–3 |
| 0 | 15 | −5 | 0–3 |
| 0 | 15 | −15 | 0–3 |
| 0 | 15 | −15 | 0–3 |

*2.4. Adhesion Test of Ice*

The diameter of the cylinder is 2 cm, the height is 2.5 cm, and the mass is 5 g. The cylinder was frozen on the coating. Frozen at −20 °C for 12 h. The sample was pushed horizontally along the surface of the coating by a dynamometer, and the maximum recorded thrust was ice adhesion force f. Ice adhesion strength is calculated as follows [18]:

$$\tau = F/A \tag{1}$$

where $\tau$ is the ice adhesion strength, F (N) is the ice adhesion force obtained from the test, and A ($m^2$) is the contact area between the cylinder and the coating surface.

*2.5. Stability Test*

The coating was placed on 800 grit sandpaper with a 150 g weight attached as a load, and the substrate was pulled at a constant speed. As the base of the coating was 2.5 cm square, two reciprocal movements (10 cm) on the base were counted as one cycle. Trials were repeated three times. In addition, ten wind tunnel tests were repeated on biochar/polypyrrole coated with three blades at −10 °C and −10 m/s. The bond test was also performed ten times.

**3. Results**

*3.1. Main Physical Property Test of Coating*

3.1.1. Fourier Transform Infrared Spectroscopy (FT-IR)

Figure 2a shows the FT-IR spectrum of biochar, with its characteristic absorption peak observed. Specifically, 3463 $cm^{-1}$ is where the stretching vibration peak of the O-H bond appears, and 2900 $cm^{-1}$ is where the stretching vibration peak of the C-H bond emerges; this group could endow the material surface with good hydrophobicity [19,20]. At 1710 $cm^{-1}$ is where the stretching vibration peak of the C=N bond is observed, and at 1286 $cm^{-1}$ is where the stretching vibration peak of the C-O bond appears. Figure 2b shows the FT-IR spectrum of the compound. The stretching vibration peak of the C-H bond is observed at 2905 $cm^{-1}$, and the stretching vibration peak of the O-H bond emerges at 3480 $cm^{-1}$; this group could endow the material surface with good hydrophobicity. The C=C bond at 1560 $cm^{-1}$ corresponds to the pyrrole ring, and the tensile vibrational peak of the C-N bond of the pyrrole ring appears at 1020 $cm^{-1}$ [21,22]. Figure 2c shows the XRD patterns of polypyrrole, biochar, and biochar polypyrrole. The XRD mode of polypyrrole has a peak at about 2θ = 26° and shows semicrystalline nature. In biochar, a peak was observed at about 2θ = 25°. The biochar/polypyrrole material exhibits two peaks at about 2θ = 25° and 2θ = 26°. Based on the SEM characterization of the coating, the XRD analyses, and the article [23,24], it can be judged that polypyrrole has been added to the biochar.

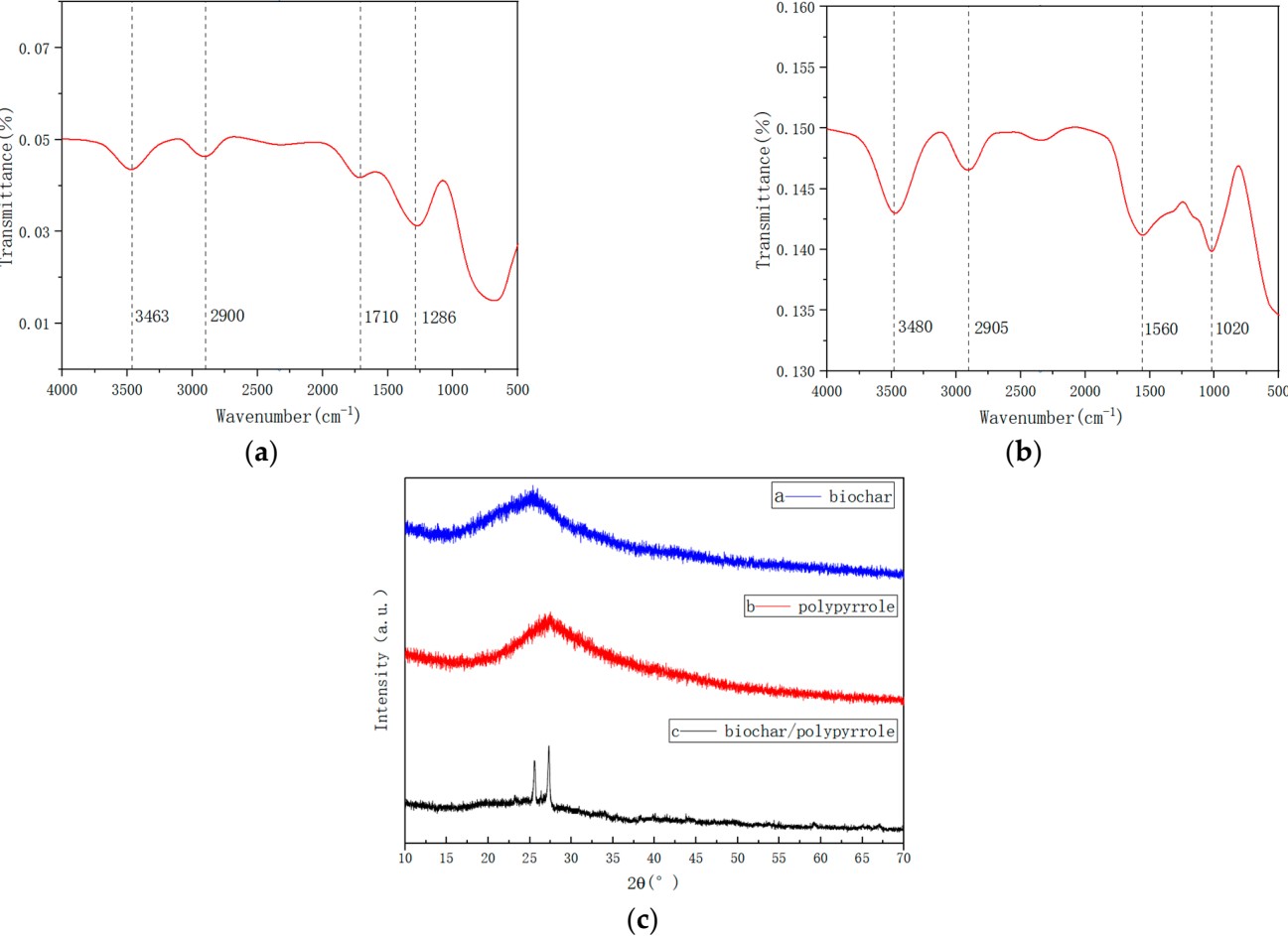

**Figure 2.** The infrared spectra of (**a**) biochar, (**b**) biochar/polypyrrole, (**c**) XRD analysis of a-biochar, b-polypyrrole, c-biochar/polypyrrole.

### 3.1.2. Morphology and Surface Analysis

Figure 3a shows the specific surface area spectrum of biochar, and Figure 3b shows the specific surface area spectrum of biochar/polypyrrole. From Figure 3a,b, it can be seen that the adsorption/desorption isotherm of $N_2$ falls into types II and III [23], indicating the multilayered surface of the material. The existence of porous structure is also indicated by the rapid rise of adsorption–desorption isotherms in the high-pressure zone. As indicated by the significant deviation between the desorption isotherm and the adsorption isotherm, there are numerous pore structures. This was confirmed by the porous interlaced structure observed under the scanning electron microscope. The specific surface area of biochar is 43.41 $m^2$/g, and that of biochar/polypyrrole is 26.17 $m^2$/g. The pore volume of biochar is 6.32 $cm^3$/g, and that of biochar/polypyrrole is 3.13 $cm^3$/g. The average pore diameter of biochar is 5.82 nm, and that of biochar/polypyrrole is 4.78 nm. These data demonstrate the large pore size and pore volume of the two materials. Due to the porous and staggered pore structure on the coating surface, the surface roughness of the coating increases. When water droplets come into contact with a solid surface, the contact area between the water droplets and the solid surface is reduced, reducing the possibility of water droplets adhering, thereby achieving hydrophobic and anti-icing performance.

In order to examine the morphology and structure of the coatings as prepared in this study, SEM equipment was employed to scan the images. The experimental voltage was set to 5 kV. Figure 4a,b show the microstructure of biochar and biochar/polypyrrole photographed.

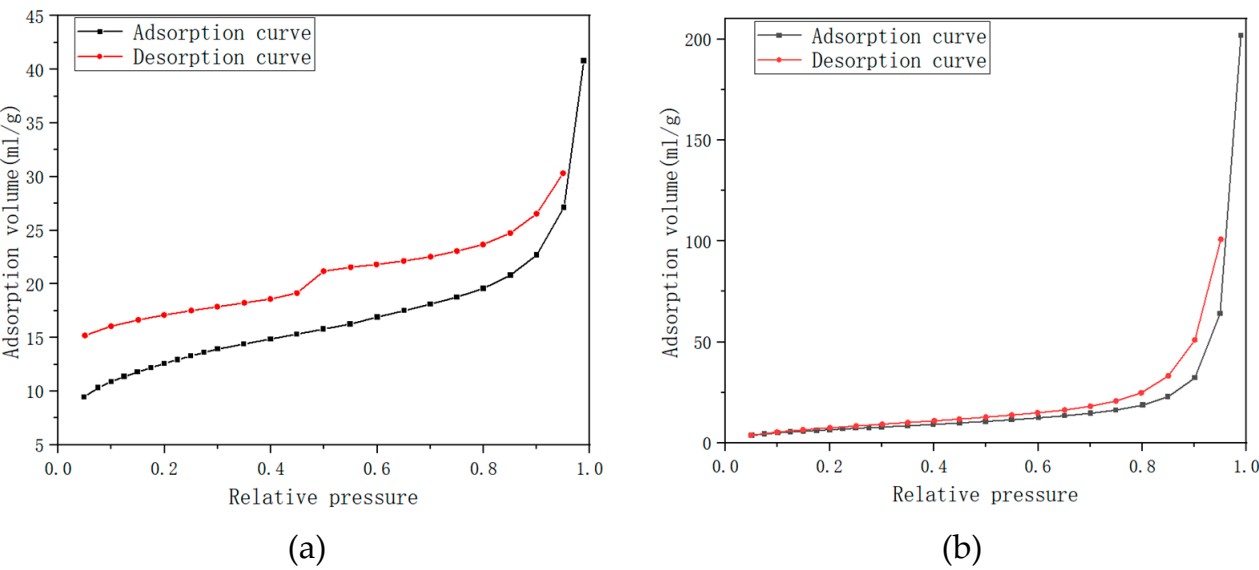

**Figure 3.** BET spectrum of (**a**) biochar and (**b**) biochar/polypyrrole.

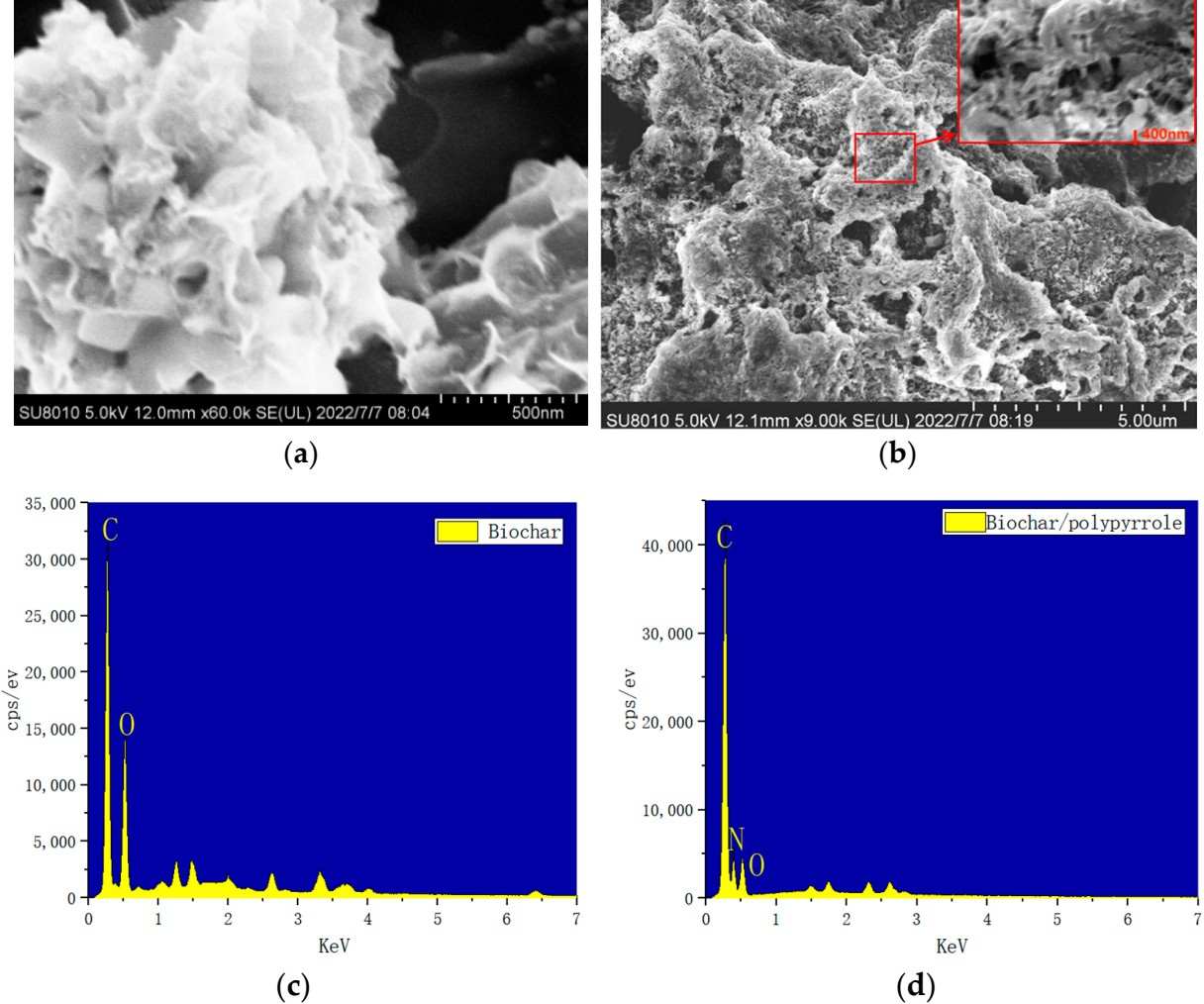

**Figure 4.** SEM images (**a**) biochar, (**b**) biochar/polypyrrole, EDS analyse, (**c**) biochar, (**d**) biochar/polypyrrole.

It can be seen from Figure 4a that the biochar material possesses a highly porous structure. The scan of the typical biochar/polypyrrole in Figure 4b shows a honeycomb-like structure as a whole. As revealed by the enlarged image, the biochar surface develops the polypyrrole nanowires with uniform morphology, in which the length of polypyrrole nanowires reaches micrometer levels [24,25]. As shown in Figure 4a,b, these pore structures have a large specific surface area and rough structures. This leads to the improvement of hydrophobicity [26]. According to the EDS analysis diagram of biochar in Figure 4c, the elemental composition is mainly composed of c and o elements. According to the EDS analysis diagram of biochar/polypyrrole in Figure 4d, the elemental composition is mainly composed of C, O, and N. Combined with the FTIR analysis in Figure 2, this is the result of polypyrrole loading on the surface of biochar.

### 3.1.3. Static Contact Angle and Sliding Angle Analysis

In order to characterize the wettability of the sample surface, a video optical contact angle measurement instrument was used to measure the static contact angle through the static drop method. When the surface of the wind turbine blade is free of any coating, the contact angle is 58° (Figure S1a), which is hydrophilic. The contact angle of a single PDMS coating is 108° (Figure S1b), consistent with previous measurements [27] showing that PDMS provides only a degree of hydrophobicity and that binding with biochar and biochar/polypyrrole is an effective strategy for coatings to produce a super-hydrophobic effect. When biochar coating is applied, the contact angle is 121° (Figure S1c), which is hydrophobic. When the biochar/polypyrrole coating is applied, the contact angle is 151° (Figure S1d), and the sliding angle is 4°, which is more hydrophobic. When there is no coating applied, these droplets diffuse to the surface. After coating, due to the rough surface of the material, the porous structure, and the large pore size and volume obtained by the BET test. So, the air is trapped in these pores. Therefore, the rough coating can be regarded as a type of biochar/polypyrrole material comprised of air and the coating trapped in pores. Thus, it can be analyzed by using the Cassie Baxter model. As it conforms to the Cassie Baxter model [28,29], the calculation can be performed through the following Equation (2):

$$\cos \theta_r = f_1 \cos \theta - f_2 \tag{2}$$

where $f_1$ and $f_2$ represent the contact fraction of water and solid on the surface of the super-hydrophobic coating and the contact fraction of water and air, respectively. ($f_1 + f_2 = 1$) $\theta$ indicates the equilibrium contact angle on the blank blade surface, and $\theta r$ denotes the contact angle between nano-coating and water. The solid coating fraction of biochar/polypyrrole composite was calculated to be about 7.7% by using the equation. That is to say, when the biochar/polypyrrole coating comes into contact with water droplets, the air interception area reaches 92.3%, with barely any water droplets reaching the gap. The water droplets on this coating are nearly spherical, showing a strong hydrophobicity. Additionally, they are easy to fall off, thus ensuring a large contact angle of the coating surface. With the biochar/polypyrrole coating reaching the super-hydrophobic state, it is easy for the water drops to fall off the surface of the test piece of biochar/polypyrrole coating, which hinders the formation of ice coating. In conclusion, the synergistic interaction of the micro and nanostructures of the coated surface and the low surface energy significantly improve the hydrophobicity of the leaf surface.

### 3.1.4. Blade Icing Bond Strength Test

In this test, the ice bond strength of glass fiber-reinforced plastic substrate blades was evaluated, the results are shown in Figure 5, and three measurements were performed and averaged. The bond strength of the blank substrate, biochar coating, and biochar/polypyrrole coating was determined as 184.68 kPa, 105.80 kPa, and 65.03 kPa, respectively. It can be found that the two types of coatings play an important role in preventing the adhesion of ice to the blade surface. The best anti-adhesion effect is produced by the biochar/polypyrrole coating, and the adhesion of the leaves with the biochar/polypyrrole

coating is weaker compared to the biochar coating and blank leaves; therefore, it can be concluded under the context of icing. When the surface of biochar/polypyrrole-coated blades freezes due to its low surface adhesion, it is easier to use external forces to remove ice from the blade surface compared to uncoated blades under icing conditions, which saves time and effort.

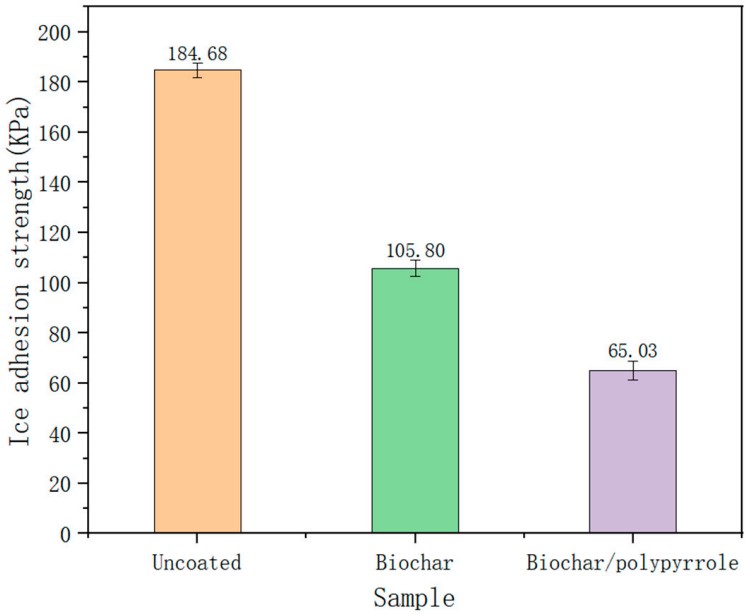

**Figure 5.** Blade icing bond strength of uncoated coating, biochar coating, and biochar/polypyrrole coating.

### 3.1.5. Stability Test

The coating was subjected to three wear tests, and the measurement revealed a change in the contact angle of the material, as shown in Figure 6a. The change in icing area and ice bond strength of biochar/polypyrrole coating is shown in Tables 4 and 5.

After three wear tests, the contact angle of the coating was changed to 132. The contact angle of biochar/polypyrrole coating was reduced by 13.8%. As shown in Figure 6b, the contact angle of the uncoated layer has changed to 52° after three wear tests, reducing by 10.3%. As shown in Figure 6c,d, the icing area of the uncoated blade surface and the ice bond strength of the uncoated coating increased by 4.2% and 5.9%. The icing area of the blade surface and the ice bond strength of the coating increased by 5.6% and 9.2%, respectively. Compare this with the uncoated blades indicating that the repeated ten tests did not cause much damage to the coating and the coating stability was good because of the rough structure of the coating after wear, but the coating still maintained its hydrophobicity.

**Table 4.** The icing area of biochar/polypyrrole coating after ten wind tunnel icing tests.

| Sample | Number of Times | Wind Speed (m/s) | Temperature (°C) | Ice Coverage Area (mm$^2$) |
|---|---|---|---|---|
| | 1 | 10 | −10 | 60.58 |
| | 2 | 10 | −10 | 59.23 |
| | 3 | 10 | −10 | 61.34 |
| | 4 | 10 | −10 | 60.85 |
| biochar/ | 5 | 10 | −10 | 61.88 |
| polypyrrole | 6 | 10 | −10 | 62.54 |
| | 7 | 10 | −10 | 64.36 |
| | 8 | 10 | −10 | 63.78 |
| | 9 | 10 | −10 | 64.66 |
| | 10 | 10 | −10 | 63.96 |

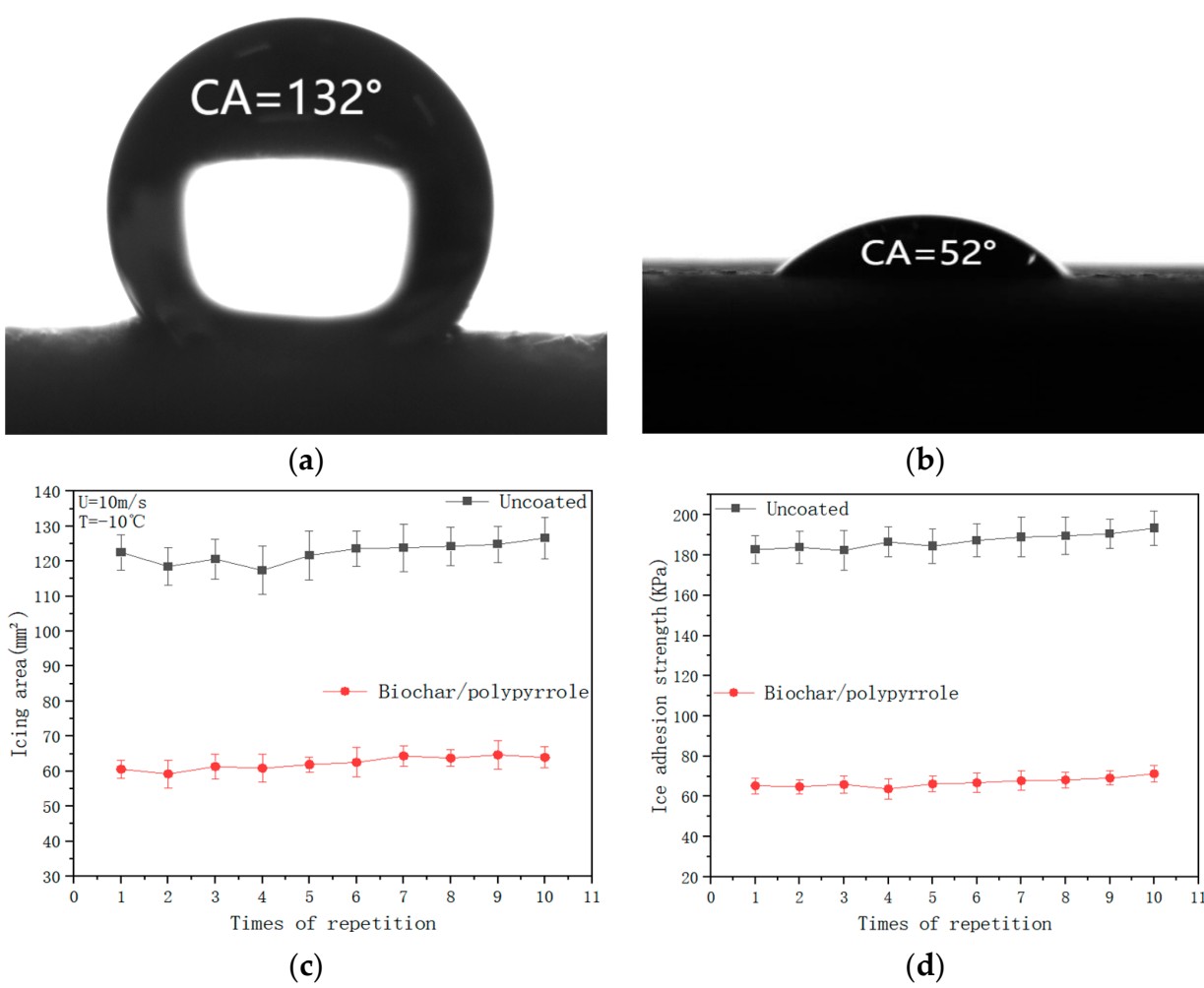

**Figure 6.** (**a**) Contact angle after wear of the biochar/polypyrrole coating. (**b**) Contact angle after wear of the uncoated coating. (**c**) Repeat ten wind tunnel test area changes. (**d**) Comparison of ten repeated ice-bond strength tests.

**Table 5.** The icing bond strength change of biochar/polypyrrole coating.

| Sample | Number of Times | Icing Bond Strength (KPa) |
|---|---|---|
|  | 1 | 65.32 |
|  | 2 | 64.89 |
|  | 3 | 65.98 |
|  | 4 | 63.78 |
| biochar/ | 5 | 66.23 |
| polypyrrole | 6 | 66.85 |
|  | 7 | 67.83 |
|  | 8 | 68.21 |
|  | 9 | 69.14 |
|  | 10 | 71.31 |

### 3.2. Anti-Icing Test of Coating

3.2.1. Icing Distribution

　　Figure 7 shows the typical test results of anti-icing coating as obtained during the wind tunnel test. Figure 8 shows the uncoated, biochar, biochar/polypyrrole icing profile captured by digitizing the photos taken in the test, which can be referenced for later quantitative analysis of the icing area.

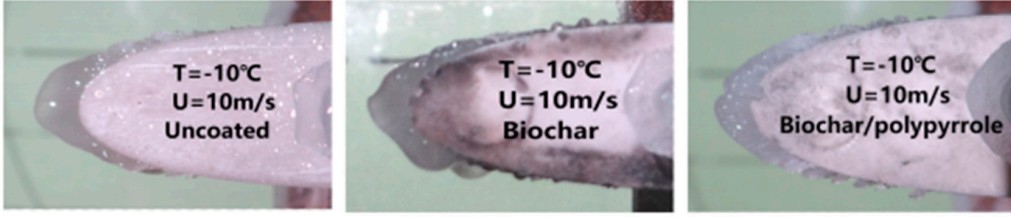

**Figure 7.** Photos of blade icing of uncoated coating, biochar coating, and biochar/polypyrrole coating at −10 °C and 10 m/s.

### 3.2.2. Icing Area

The experiment was repeated three times and averaged. As shown in Figure 9, the test data are shown in Table 6 above test data; at 0–3 min, the amount of ice formed on the blank blade surface increases significantly over wind speed. The amount of ice formed on the surface of the two types of blades after the coating is reduced significantly compared to the uncoated blade surface, and the amount of ice formed on the surface of the biochar/polypyrrole-coated blade is smaller compared to the biochar-coated blade surface. Therefore, both biochar and biochar/polypyrrole can produce an anti-icing effect for the blades, of which biochar/polypyrrole performs best in this respect.

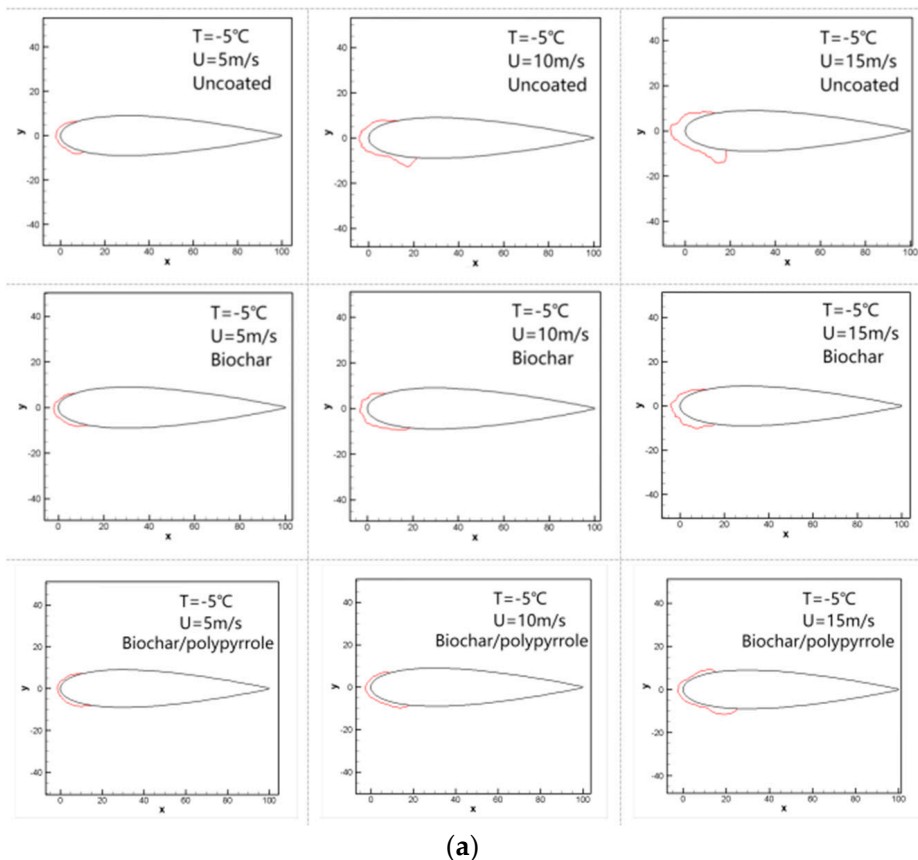

(**a**)

**Figure 8.** *Cont.*

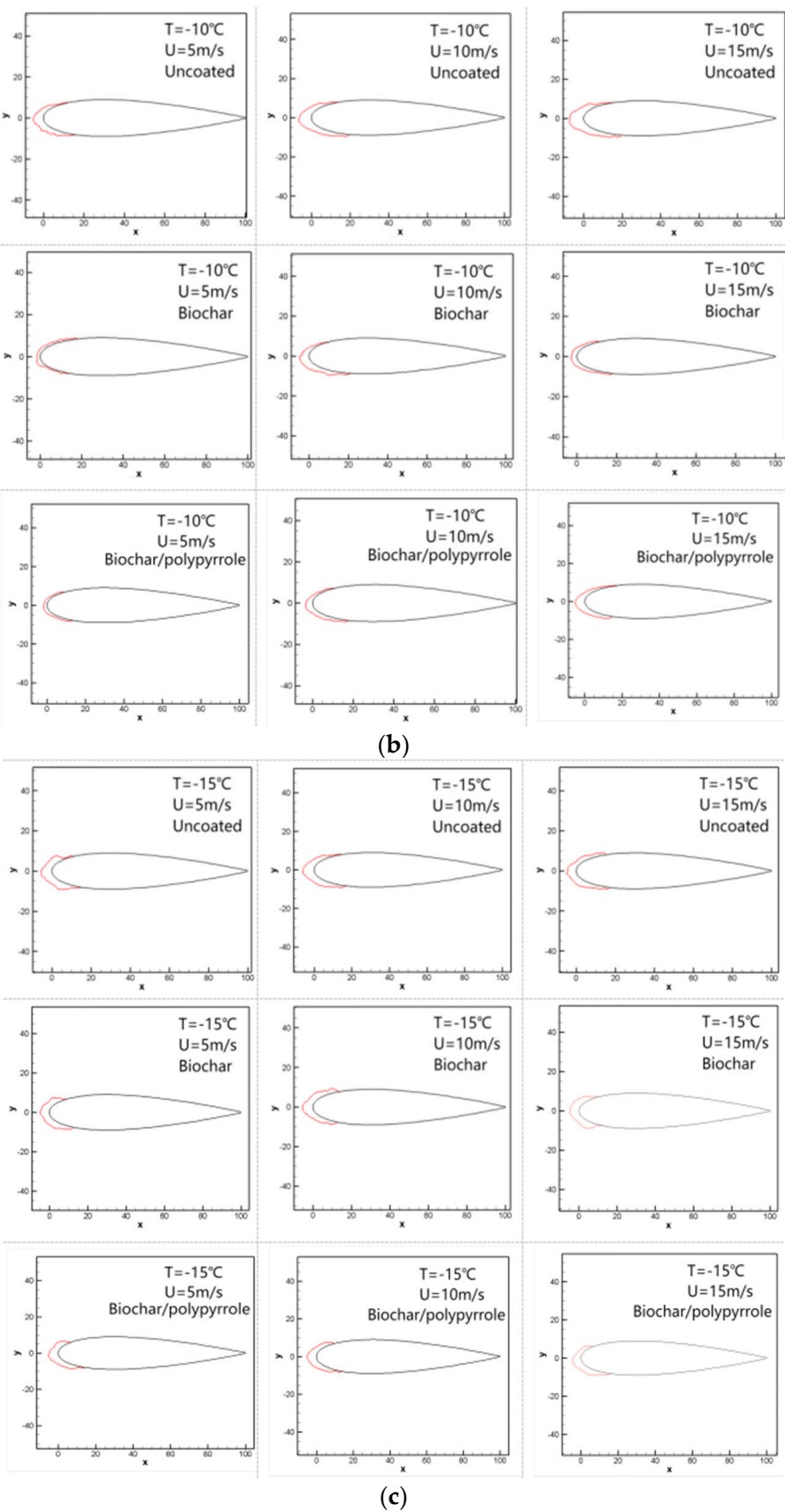

**Figure 8.** The maximum icing area of uncoated coating, biochar coating, and biochar/polypyrrole coating at (**a**) −5 °C. (**b**) −10 °C. (**c**) −15 °C.

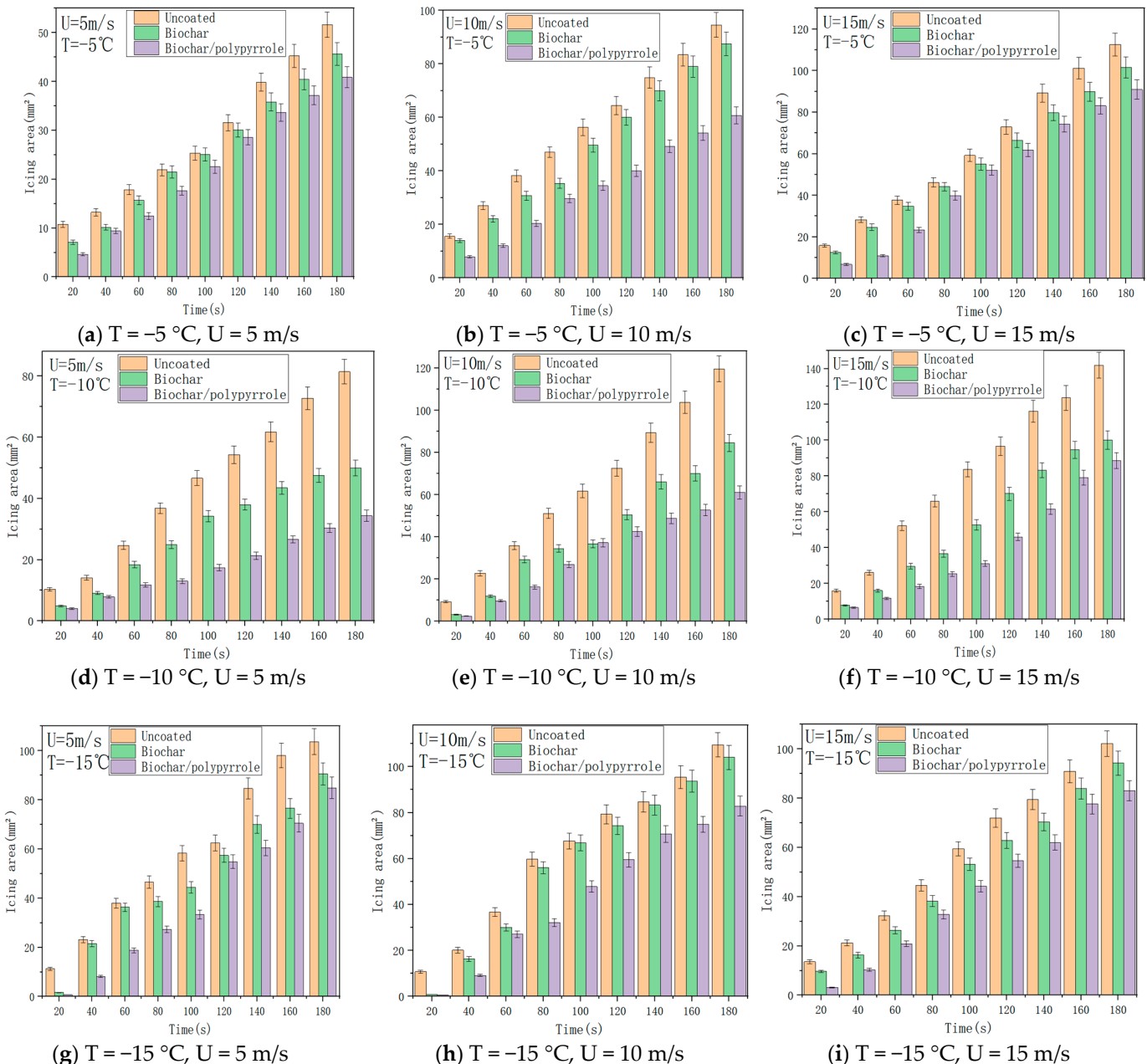

**Figure 9.** Icing area of uncoated coating, biochar coating, and biochar/polypyrrole coating.

**Table 6.** The maximum ice coverage area of uncoated, biochar, and biochar/polypyrrole.

| Sample | Wind Speed (m/s) | Temperature (°C) | Maximum Ice Coverage Area (mm²) |
|---|---|---|---|
| uncoated | 5 | −5 | 51.65 |
| | 15 | −5 | 112.52 |
| | 15 | −10 | 141.86 |
| biochar | 5 | −5 | 45.64 |
| | 15 | −5 | 101.52 |
| | 15 | −10 | 100.16 |
| biochar/ polypyrrole | 5 | −5 | 40.88 |
| | 15 | −5 | 90.92 |
| | 15 | −10 | 88.60 |

### 3.2.3. Maximum Icing Thickness of Leading Edge

Three experiments were performed, and the data from the three experiments were averaged to give Figure 10. The maximum thickness of blade icing is obtained through the ice-type image of the blade as captured from the anti-icing experiment conducted in the low-temperature wind tunnel. Additionally, the maximum thickness of ice formed on the surface was measured at the position of the thickest icing on the leading edge of the three blades, as shown in Figure 10 and Table 7. In identical working conditions, the main icing area on the blade is the leading edge of the blade. In terms of the maximum icing thickness, the uncoated blade is the highest, followed by the biochar-coated blade and the biochar/polypyrrole-coated blade in order. It can be seen from above that biochar/polypyrrole coating performs well in the anti-icing effect.

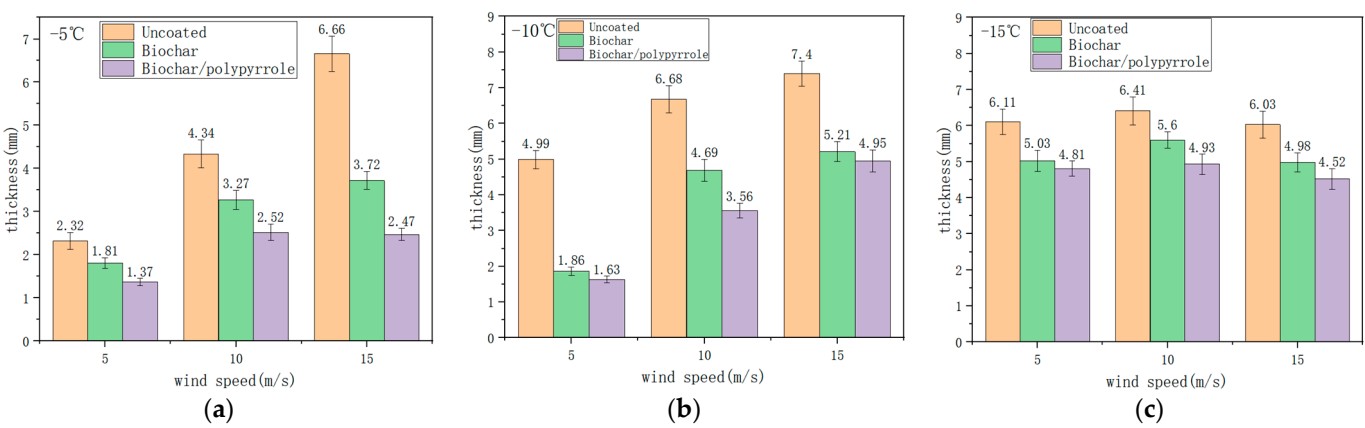

**Figure 10.** Comparison of the maximum icing thickness at the leading edge of the blade (**a**) at −5 °C; (**b**) at −10 °C, and (**c**) at −15 °C.

**Table 7.** The maximum ice thickness of uncoated biochar and biochar/polypyrrole.

| Sample | Wind Speed (m/s) | Temperature (°C) | Maximum Ice Thickness (mm) |
|---|---|---|---|
| uncoated | 15 | −5 | 6.66 |
| | 15 | −10 | 7.40 |
| | 15 | −15 | 6.03 |
| biochar | 15 | −5 | 3.72 |
| | 15 | −10 | 5.21 |
| | 15 | −15 | 4.92 |
| biochar/ polypyrrole | 15 | −5 | 2.47 |
| | 15 | −10 | 4.95 |
| | 15 | −15 | 4.52 |

## 4. Discussion

This paper adopts the wind tunnel test method, a wind tunnel test to simulate natural wind speed and wet rain environment, records the icing distribution on the blade surface, and analyzes the anti-ice effect of the coating. It is better than the static anti-ice test, the test method of static anti-ice test for Chen et al. and Zhang et al. [30]. Because wind tunnel tests are used for anti-ice research, they can reflect the natural environment of the ice conditions close to reality.

In this paper, a wind tunnel test method is used to simulate the natural wind speed and wet rain environment to record the ice distribution on the blade surface and analyze the anti-icing effect of the coating. Since the wind tunnel test is used for anti-icing studies, it is superior to the static anti-icing test done by Chen et al. [30]. Because the wind tunnel

test is used for anti-icing studies, it reflects the ice conditions in the natural environment and is close to reality.

Valentini et al. [31] studied the icephobicity of graphene surfaces functionalized by sodium ions, chloride ions, or methane molecules investigated using molecular dynamics simulations. The icephobicity of the surface is proven by the freezing temperature. Rezazad also used waste as raw materials to prepare biomass carbon materials [32], but his preparation process is more complex. The contact angle of polytetrafluoroethylene/heat-shrinkable polyvinyl chloride material is 150°, Jiang et al. [33]. Although there is little difference in the contact angle and hydrophobicity of the two, there are many kinds of materials required in the preparation process, and the requirements for technology and energy consumption are large. Therefore, carbon nanomaterials with super-hydrophobic and super-hydrophobic properties have good anti-icing effects. However, the high synthesis cost of these materials limits their applications. Compared with these methods, this paper uses cattle manure biogas residue biochar as the material base, which can shorten the production time of the coating and is green and efficient, in line with the theme of effective utilization of agricultural waste [34].

Biochar polypyrrole coatings froze significantly less than uncoated leaves at nine different wind speeds and temperatures. With a wind speed of 5 m/s and temperature of −10 °C, the icing area and maximum icing thickness decreased by 55.7% and 41.90%, respectively, compared with uncoated blades. At a wind speed of 5 m/s and a temperature of −5 °C, the water rolled on the blade surface before freezing on the coating, confirming the above results. When the blades were at lower temperatures, the coated blades still showed some ice resistance effect, reducing the maximum icing area and thickness by 20.85% and 62.9% compared with the uncoated blades, respectively, which is better than previous studies [35]. This is because the hydrophobicity of the coating causes the droplets to roll before freezing due to gravity. Due to biochar/polypyrrole having high porosity and rough micro-nano grading structure, it can make the material and water contact area greatly reduced, reducing the adhesion of the ice, which is important for using external mechanical forces to peel the ice easily. Among other things, excellent mechanical properties, wear resistance, and repairability can also effectively solve the problem of blade mechanical damage [36].

Liu et al. [18] used the same ice bonding force measurement method and found that the bonding force of MC/$Fe_3O_4$ coating was higher at 68.34 KPa than 65.03 KPa of biochar/polypyrrole coating, showing good performance in the bond strength of the coating. Bharathidasan et al. [37] have studied the ice-adhesion strengths of coatings with different wettabilities, namely hydrophilic polyurethane and PMMA, hydrophobic silicones, and super-hydrophobic silicone and PMMA-based nanocomposite coatings. The wettability of the surfaces was evaluated by measuring the water contact angle (WCA) and sliding angle. A custom-built instrument based on the zero-degree cone method was used for evaluating the ice adhesion strength of these surfaces. In a comprehensive analysis, the measurement method is relatively complex and inconvenient to operate. In this article, a thrust meter is used for measurement, which is simple to operate and convenient for data recording. The biochar/polypyrrole coating affects the bond strength of ice and is beneficial to the de-icing of the blade surface by external forces. In addition, the micro/nanoscale air pockets trapped in the hierarchical structure of the super-hydrophobic surface act as a thermal barrier to cut off the heat transfer during the icing process and hence, effectively reduce the ice adhesion capacity [38,39].

## 5. Conclusions

The effects of the hydrophobic coating on the anti-icing performance of wind turbine blades were explored. The following conclusions are drawn:

(1) The contact angle of biochar is 121°, and the contact angle of biochar/polypyrrole is 151°. The biochar/polypyrrole coating performs better in the anti-icing effect.

(2) The main icing area of the blade is located at the leading edge of the blade, the thickness of icing on the uncoated blade is greater compared to the biochar-coated blade, and the thickness of icing on the biochar/polypyrrole coating is the smallest.

(3) The adhesion of coated blades is not as strong as that of uncoated blades. It is 105.80 kPa for biochar coating and 65.03 kPa for biochar/polypyrrole coating. This plays a significant role in preventing the blade surface from icing and de-icing.

**Supplementary Materials:** The following supporting information can be downloaded at: https://www.mdpi.com/article/10.3390/coatings13040759/s1, Figure S1: contact angle and water drop shape of (a) uncoated blade surface; (b) PDMS coating; (c) biochar coating; (d) biochar/polypyrrole coating.

**Author Contributions:** Conceptualization, X.L. (Xiaoheng Li)., Y.L. and F.F.; formal analysis, X.L. (Xiaoheng Li), Y.L. and F.F.; funding acquisition, F.F.; investigation, X.L. (Xiaoheng Li) and Z.M.; methodology, X.L. (Xiaoheng Li), X.L. (Xiaojuan Li) and Z.M.; supervision, Y.L. and F.F.; validation, Z.M. and Y.L.; writing—original draft, X.L. (Xiaoheng Li); writing—review and editing, X.L. (Xiaoheng Li), Y.L. and F.F. All authors have read and agreed to the published version of the manuscript.

**Funding:** This work was supported by Project Grant NO.52076035 supported by the National Natural Science Foundation of China (NSFC), and provided by Project Grant No. IADL20200405 supported by The Open Fund of Key Laboratory of Icingand An-Ti/De-Icing, China Aerodynamics Research and Development Center.

**Institutional Review Board Statement:** Not applicable.

**Informed Consent Statement:** Not applicable.

**Data Availability Statement:** Not applicable.

**Conflicts of Interest:** The authors declare no conflict of interest.

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
