# Peer review of "An Experimental Study on Biochar/Polypyrrole Coating for Blade Anti-Icing of Wind Turbines"

_coatings, doi:10.3390/coatings13040759_

Round 1

Reviewer 1 Report (New Reviewer)

Research in the field of anti-icing coatings is extremely relevant. The article has a practical focus and has a number of comments that need to be addressed:

1) The last paragraph of the introduction is more like a conclusion. Here it is necessary to reflect the scientific novelty, the idea that the authors propose and clearly formulate the purpose of the work.

2) L 41. Remove the term nanocoatings.

3) Clause 2.2. Test methods should be described in detail.

4) Clause 2.3. What is the mass of the cylinder?

5) The authors wrote that the chemical interaction is visible from the IR spectra, but this is not visible. What interaction are the authors talking about? It is necessary to apply the scheme of polymerization, fixing the polymer on the surface of the biochar and fixing the composition on the surface of the blade.

I don't understand the principle of choosing pyrrole, why would it lead to hydrophobicity? What is the surface free energy of polypyrrole?

6) L. 184 The authors wrote that an increase in pore size improves anti-icing properties, it is not clear why.

7) How did the authors measure the roll angle? I have serious doubts that 151 degree contact angle coatings have such low roll angles. Please provide a photo or video of the experiment.

8) Fig. 5 does not make any sense, if you want to leave it, then transfer it to additional materials.

9) L. 251 What does unused mean?

10) L. 252-254 It is not clear what the authors wanted to say.

11) Fig.7 It is necessary to make a comparison with the original substrate.

12) Shorten the conclusions and write briefly, in fact, the results obtained.

13) Section 4 contains only a comparison (this is of course good), where is the discussion?

14) Format the text as required.

Author Response

Reviewer 2 Report (New Reviewer)

The presented manuscript is well-written and well-organized and can be accepted for publication after the corrections listed below.

1. Fig. 4 - please, add elemental mapping in elemental contrast here, as well as discuss the distribution of elements within the text of the manuscript.

2. It looks like the infrared spectra are not a trustworthy source of truth regarding phase composition. I want to propose using XRD studies instead, of course, if the authors can not perform TEM analysis. 

3. As for the inclination angle - please, provide more detail regarding measurement errors, how did the authors perform such measurements and define possible errors?

4. The review of recently published articles can be improved a little. For example, the article [1] can be potentially interesting for the authors of the manuscript. 

References

[1] doi.org.10.1021/acsami.2c22780

Round 2

Reviewer 1 Report (New Reviewer)

The authors did a great job, wrote detailed answers, the article has become better, but there are points that need to be discussed in more detail:

1)      Maybe you misunderstood me, but the last paragraph of the introduction should contain a clearly formulated goal of the work and novelty. Please write what is the idea of the work, while this idea should be traced from the introduction to the conclusions.

2)       For the first time introduced abbreviations must be deciphered.

3)       L127 "legal sample" - what does it mean?

4)       Research methods should be described in more detail. L136 What does "suspension method" mean?

5)      "It is necessary to apply the scheme of polymerization, fixing the polymer on the surface of the biochar and fixing the composition on the surface of the blade." The authors have added a description of the deposition method, but the chemistry of the process should be shown. Provide a scheme of polymerization and fixation on the surface of the substrate, which will be understandable to the reader.

6)      The choice of pyrrole: if you write that the free energy of the surface of polypyrrole is from 30 to 145 mJ / mm2, then this is a lot! Above 30 mJ/mm2 SFE you will not get stable super hydrophobic properties. This raises the question of choosing a hydrophobic agent.

7)      The roll angle measurement methodology is not reproducible. Look at the literature: what parameters need to be controlled (volume of a drop, the rate of inclination of the sample to the horizon, etc.). What are the statistics of this experiment?

Round 3

Reviewer 1 Report (New Reviewer)

The authors did a good job of improving the text, the article became noticeably better. May be recommended for publication. I still have some small questions, but the decision will be up to the journal editor:1)      For clarity, it would be worthwhile to give a scheme for obtaining a modifier and fixing it on the surface. This would allow the reader to see the chemistry of the process.

2) “Thank you for your careful work. The author is sorry for the error in expression, the surface energy 30-145 mJ / mm2 is not the surface energy of the material prepared by the author is this, we know that polypyrrole has high surface energy, but check the relevant literature, doping surfactants, Hydrophobic polypyrrole materials can also be prepared by doping with low surface energy substances. Surfactants as low surface energy substances can reduce the surface tension and surface energy of substances. These low surface energy substances are often used as dopants, cetyl trimethyl ammonium bromide is a surface activator can come to reduce the surface energy of polypyrrole."

It remains unclear to me what reduces the surface energy? In the articles cited by the authors, CTAB with a long hydrocarbon substituent was used, but there is no such modifier in this article.

3) The article is hard to read, I am not a native speaker, but I recommended that the article be submitted for English corrections.

Author Response

This manuscript is a resubmission of an earlier submission. The following is a list of the peer review reports and author responses from that submission.

Round 1

Reviewer 1 Report

The work entitled „An Experimental Study on Biochar/Polypyrrole Coating for Blade Anti-icing of Wind Turbines” describes the possibility of application of biochar/polypyrrole layers as a passive anti-acing coating.

The general idea of the work seems interesting for the readers of the Journal, especially taking into account the practical approach of this work. The novelty of the approach is rather clear. However, there are some major issues in the methodology and the data analysis that need clarification before further reconsideration:

1.       No information about the purity of the reagents used is given.

2.       The quality of figures is rather poor.

3.       In FTIR: a) the y axis should be labeled, b) why background was not subtracted? C) please double-check the assignment of the signals, d) why no biochar signals are observed in the spectrum of polypyrrole/biochar?

4.       SEM: since the images of different magnification are shown it is hard to compare them and draw any conclusions.

5.       What is the result of contact angle measurements and anti-acing of pure polypyrrole.

6.       Generally text is well written, but there are some strange sentences, like: “plenty of functional groups”, “sem diagram”. All in the text not only a/b of particular figures should be given, but also its number.

Reviewer 2 Report

The manuscript "An Experimental Study on Biochar/Polypyrrole Coating for 2 Blade Anti-icing of Wind Turbines" could represent an interesting study about the icephobic materials however the proposed coating should be better characterized and the anti-icing property of the superhydrophobic surface must be compared with the recent literature. Please see the attachment for further revision.  

Reviewer 3 Report

The topic is important and the windtunnel experiments are a proper way to address the topic. However, the experiments and reporting are inadequate to convince the reader.

My main concerns are:

Are the results shown in Figs 8-10 from single experiments or multiple experiments? There are no error bars, and no idea how many data points are included. On page 10, line 273, error ranges are mentioned, but no quantative data on errors is given.

The above question is important because durability/robustness of superhydrophobic coatings is based on micro/nanostructures, and their survival under icing conditions is not discussed at all. If the experinents are all done on virgin samples, I agree that they work nice, but do they work nice repeatedly? Ten cycles ? Hundred cycles? Windmill lifetime? What would be the reapplication frequency of the coatings? SEMs and contact angle data of blades after icing tests should be shown.

The way ice adhesion measurements are carried out is not decribed at all, and this is non-trivial measurement.

I think that too much emphasis is given to FTIR and BET analysis, because they are not really used to explain the icing results.

Because PDMS is used in the mixture, and PDMS is known to be highly hydrophobic, might it be that the results are due to PDMS, and not at all due to biochar/pyrrole?

The biggest failure of the manuscript is that it does not cite scientific literature. In the results and discussion section the new results of this work must be compared to older works. What contact angles did other researchers measure for their coatings? Why are the new ones different/better? What ice adhesion results were achieved in older works? Why are the present ones better/different? How much ice accumulated on blades in published works? Were they conducted the same way as the current ones? The readers want to know how the current work compares with old ones, so that she can evaluate the value of the new work. It also helps authors to write claims: they should bring the novel aspect to fore; they should highlight their results which are better than old ones; they should explain the mechanism why their coating is superior to old ones.

Technical comments:

General comment: figure captions are non-informative. The caption must tell the reader what is important in the figure, what should the reader pay attention to. It is not enough to name a figure.

P2, L86: TS set to 7% is not clear

P3, L113: MVD: volume diameter is oxymoron: unit of volume is cubic meter, unit of diameter is meter.

Table 3: Angle (not angel) of attack, and time are constants. Table misleadingly suggest that 6 experiments were done. Why not make it look like a 3x3 matrix?

P5, L158: "polypyrrole is grown on the surface of biochar" This is very unclear, because they were thoroughly mixed on P2, L94.

I do not believe that the decimals in contact angle measurements are real: especially for high CAs the measurement is increasingly inaccurate, and I would adopt to report CAs simply as 58, 121 and 152 degrees.

P6L197: "coating value f1". Standard terminology would be: solid fraction

P6L201: large contact angle does not mean that the surface is superhydrophobic !!! Low sliding angle must also be present.

P7L225: Fig. 7 is claimed to show anti-icing coatings. It shows icing results.

Units of pressure: kPa, not kpa

Figure 8: I find this unnecessary. Maybe 3 frames is enough to show some major trends.

Text above Figure 9: I would just describe major trends found, without numbers. The numbers are in the figure.

P9, L250: "increases significantly over time" but the experiments has constant time of 3 minutes according to table 3.

Figure 9: curiously, the coatings seem to work best at -10oC ??!!

P10L295, ice type: Yes, it would be interesting is ice type was examined. It may be that at -10oC ice is somehow different.

In conclusion the authors are free to speculate a bit, but I find if too bold to claim that this coating is low cost. Cow dung is cheap, but PDMS is pretty expensive, pyrrole monomer is not cheap either.  

Round 2

Reviewer 1 Report

Authors have answered all the questions asked. I would suggest acceptance at this point.

Reviewer 2 Report

the revisions have been done and the manuscript is suitable for a publication

Author Response

Thank you for your approval.

Reviewer 3 Report

I was unable to appreciate the results because materials preparation section was inadequate: it is not clear what amounts of biochar and PDMS went into the mixture, and therefore, I suspect that the major hydrophobic effect comes in fact from PDMS.

It is uclear how many experiments have been conducted: it states on page 9 line 250 that the amount of ice formed increases significantly over time, but in Table 3 icing time is constant. Figs. 8 and 9 contain a wealth of infromation, but despite claim "given the error range" (page 10, 274), the number of repeats is not evident.

Durability of the coating is not studied properly. It is fairly easy to obtain superhydrphobic and low adhesion surfaces, and the real question is if they survive the intended environment in the long run.

The manuscript fails in the discussion part. Discussion in supposed to weave the new results into the fabric of science. The authors have to compare their new results in the light of old results, and explain what is really new, what is minor improvement, what new experimental and measurement techniques were used, what is a new application of known principle or technique, or old principle and technique applied to new materials etc. There were no references in Results and discussion part. What CAs did previous workers achieve ? What ice adhesion has been measured before? How does the current manuscript differ or improve from old works? What explains the differences? What are the benefits of the proposed new method? "Great potential to reduce costs" needs more explanation: cow dung is cheap, but considering pyrrole, PDMS, hexane, pyrolysis etc. is this really different from the old ones?
